# Weak Fiber Bragg Grating Array-Based In Situ Flow and Defects Monitoring During the Vacuum-Assisted Resin Infusion Process

**DOI:** 10.3390/s24237637

**Published:** 2024-11-29

**Authors:** Xiao Liu, Zuoyin Tang, Xin Gui, Wenchang Yin, Jingyi Cao, Zhigang Fang, Zhengying Li

**Affiliations:** 1School of Information Engineering, Wuhan University of Technology, Wuhan 430070, China; liuxiao001@whut.edu.cn; 2School of Materials Science and Engineering, Wuhan University of Technology, Wuhan 430070, China; 3National Engineering Research Center of Fiber Optic Sensing Technology and Networks, Wuhan University of Technology, Wuhan 430070, China; 4Unit 92228 of the Chinese People’s Liberation Army, Beijing 100072, China

**Keywords:** fiber Bragg grating array, three-dimensional, flow monitoring, defects

## Abstract

Monitoring of real-time flow and defects in the vacuum-assisted resin infusion (VARI) process can provide important guidelines for full impregnation of dry reinforcement. A weak fiber Bragg grating array was employed to obtain quasi-distributed monitoring results in real-time. Sensitivity testing of different kinds of coated optical fiber sensors (OFs) was carried out first, and the polyacrylate-coated OF showed a greater wavelength-shift response than the polyimide-coated one. Then, two- and three-dimensional flow monitoring tests were carried out. During the resin-filling stage, three trends of strain curve were identpified in relation to the different placement setups of embedded OFs, the resin flow direction, and the different vacuum-bagging methods. The monitoring criteria were analyzed and the results were compared with the visual inspection, showing good agreement and indicating the ability of the fiber Bragg grating array. Finally, defects including dry spots and voids were introduced and reflected in the maximum changed strains of FBGs due to the smaller stress relaxation, indicating the potential to characterize the local flow state and permeabilities experimentally based on these quasi-distributed sensing methods.

## 1. Introduction

Vacuum-assisted resin infusion (VARI) process is a member of the liquid composite molding (LCM) family and a cost-competitive composite manufacturing process involving out-of-autoclave techniques, which is widely used in naval ships, submarines, and maritime structures [1,2]. In the resin transfer stage, the viscous resin gradually fills all spaces on the one side of the mold, which exist between the fiber mats, between the fiber tows, and within the fiber tows due to the pressure gradient caused by the vacuum drawing on the system [3]. The drawback of VARI and other LCM technologies is that a new production variable needs to be considered: the optimization of the resin flow to properly wet all the fibers [4]. Insufficient resin impregnation produces defects like dry spots in places where the resin cannot reach or flow, which can seriously degrade the quality of finished product and jeopardize the mechanical integrity of structure [5]. However, it has been found that the resin flow during injection can be rather unpredictable, probably because of the unavoidable rearrangement of fibers and associated race-tracking effects, especially for large-size and thick composite structures, as the resin flow in the thickness direction cannot be ignored [6]. Situations of “demand over capacity and engineering before theory” have emerged in the VARI process. In situ monitoring technologies are more intuitive and effective and should in time be developed accordingly to adjust important process parameters such as pressure and the location of vents to reduce defects.

Many researchers are working on methods for resin flow monitoring based on different physical principles. According to the relative relation between the sensor and the fiber preform, monitoring methods can be divided into two kinds of methods: non-invasive and invasive. Among the non-invasive methods, the most cost-effective and convenient monitoring method is to use a transparent mold or vacuum bag to compact the fiber preform and track the flow front visually. Yun et al. [7] conducted 3D radial flow experiments using two cameras placed on the top and bottom of the fiber preform to record the advance of the flow front and proposed a non-invasive method to determine the permeability tensor. Bin et al. [8] used real-time images and MATLAB code to analyze the evolution of resin flow fronts. Darcy’s law had been applied to obtain the local and global in-plane permeability results, which were in good agreement with those obtained by the radial permeability experiments. Although visually tracking has the advantages of simple operation, low cost, and little effect on the sample, it requires a transparent mold of low stiffness, which can easily lead to large bending deformation due to the compaction pressure. The quality of the product is thereby compromised and the visually tracking method is therefore not applicable in real manufacturing scenarios.

In other non-invasive methods using ultrasonic longitudinal waves, there are assumptions that the respective wave velocities are constant in the dry and the fully saturated fabrics. Then, the flow front along the thickness direction is linearly correlated with the flight of time of the ultrasonic longitudinal wave. Becker et al. [9] and Konstantopoulos et al. [10] monitored flow fronts in the thickness direction based on ultrasonic sensors, and the unsaturated out-plane permeability results for carbon and glass fabrics at three different ratios of fiber volume showed good compliance with relative deviation. Textile-induced inhomogeneities and varying measurement parameters, e.g., injection pressure, were the main reasons for the deviations. With non-invasive measurement methods, errors or even mistakes can be introduced, as the real-time flow fronts are often acquired indirectly based on assumptions.

Pressure, temperature, electrical impedance, and even optical signals change before and after the resin reaches the dry fiber [11,12]. Fratta et al. [13] proposed a method for monitoring resin flow based on a combination of a few pressure sensors and numerical simulation; the results showed good agreement between the real flow-front profiles and the estimated ones at different time steps, with adequate detection of small flow variations. Pressure sensors are commonly used for 2D in-plane flow monitoring due to the need for sufficient monitoring range, considering limitations of low accuracy. Regarding thermal sensors, Tuncol et al. [14] used thermocouple sensors to monitor a resin transfer molding process. Thermal sensors are low-cost and durable, but there faced many limitations such as high resin flow rate, the high specific heat of resin, and low accuracy. So, thermocouples should not be preferred over sensors of other types.

For electrical sensors, the monitoring mechanism of the flow front inside the fiber preform results in changes in the electrical impedance around the sensor before and after the flow front arrives. Muhammad et al. [15] used graphene-coated piezo-resistive fabrics and monitored the unidirectional flow during the VARI process, based on changes in resistance. Zhang et al. [16] designed a flexible Pt-coated film capacitive sensor; the flow front, thickness variation, and curing inside composite preform could be captured based on the variation of the capacitance curve and its second derivative. Tifkitsis et al. [17] used a lineal dielectric sensor composed of two uniformly twisted insulated copper wires to track resin flow fronts. However, when electrical sensors are used in conjunction with the primary object (carbon fiber) in the manufacturing of high-specification composite components, there is a high risk of electric field interference issues occurring. The conductive carbon fiber can also provide new self-sensing monitoring methods. José et al. [18] and Kim et al. [19] developed novel resistive carbon-fiber-reinforced sensors. The output voltage or resistance of carbon fiber tows can be used to monitor resin flow fronts and realize self-sensing and self-monitoring for composite manufacturing processes. However, electrical sensors are normally self-made, with the disadvantages of lack of design standard and limitations in types of reinforcing fibers, making their promotion and application difficult.

Due to their lightweight, small size, compatibility with conductive fibers, and the ability to conduct multi-point measurements through wavelength division multiplexing (WDM) technology, optical fiber sensors (OFs) are ideal invasive flow-front monitoring sensors and have attracted the attention of many researchers. As the resin reaches the etched region of the OF, the total reflection condition is disrupted, resulting in a decrease in optical intensity signals. Lim et al. [20] used etched OFs to detect resin flow fronts in cladding sections. Liu et al. [21] employed etched OFs to monitor the three-dimensional flow in stacks of biaxial E-glass fiber non-crimp fabric (NCF) and unidirectional carbon fiber layers, and the monitoring ability of etched OFs for different kinds of fabrics was verified. The etched OFs normally included no more than four measurement points on a single OF, due to the fragility of the etched region without coating and the loss of signal with many etched regions.

Similarly, as the resin reaches the fiber-end/air interface, the photodetector records a significant dropping of the reflected signal because the mismatch between the refractive index of the resin and the fiber optic is very low. Therefore, OFs based on the Fresnel reflection law can be used to detect resin flow fronts. Antonucci et al. [22] continuously monitored the reflected light signals at the end of the optical fiber and obtained the resin flow fronts in the thickness direction. He et al. [23] used a fiber optic micro-flow sensor with super-infiltration coating to monitor the micro-flow of resin between reinforced fibers. They found that the proportion of the effective resin flow time compared with the usually estimated flow time was as low as 54%, and the influence of the flow time parameter on void development was more noticeable than that of heating-up rate. However, a Fresnel reflection OF is a point sensor of one single OF.

The strain or temperature changes caused by the arrival of the resin can also be sensed by conventional fiber Bragg grating (FBG) sensors, making them suitable as local sensors for monitoring flow fronts. Eum et al. [24] used temperature changes in FBG sensors and strain changes in long strain gauges to monitor resin flow fronts. Gupta and Sundaram [25] employed three configurations of OF: etched OFs, embedded FBGs, and non-embedded FBGs to monitor flow fronts. The results showed the multi-location sensing capability of OFs with WDM technology, and the information about the time of arrival of the resin flow front provided valuable inputs for analytical modeling for complex structures. Yu et al. [26] embedded multiple FBG sensors and piezoelectric sensors in different layers to form a hybrid piezoelectric–fiber sensor network, and the measurements were combined to complement each other for estimation of the 3D resin flow front. Experimental results confirmed that the hybrid piezoelectric–fiber sensor network was capable of successfully estimating the 3D flow front in situ. However, conventional FBG sensors usually need to strip the coating, inscribe gratings, and perform the secondary coating process, which weaken the mechanical properties of the OF and limits the number of sensing points. Weak fiber Bragg gratings fabricated inline on a draw tower can overcome these drawbacks [27,28].

There have also been many studies about real-time monitoring in composite manufacturing processes using disturbed optical sensors including Brillouin scattering or optical frequency domain reflectometry. Most of these have focused on the cure-induced strain monitoring of CFRP/CFRTP [29,30,31], with only little research on resin flow monitoring [32]. In summary, each sensor has its own advantages and disadvantages, which are summarized in Table 1. In result, the lack of quasi-distributed flow front monitoring technology during manufacturing makes it difficult to characterize the local permeability and reconstruct the three-dimensional flow front under the complexity and non-uniform characteristics of the fiber preform [33]. The novelty of our paper is the presentation of a detailed spectrum and the strain responses obtained using a weak FBG array during the resin-filling stage of the VARI process. Different strain curves with different setups are discussed, and the resin flow front and manufacturing defects were monitored using the weak FBG array.

## 2. Flow Monitoring Theories and the Weak FBG Array

### 2.1. Compressibility of Dry and Wet Fibrous Reinforcements

For the LCM process family, the compaction of the dry and wet fabric reinforcement inside a mold or vacuum bag is vital for determining the mechanical properties of the finished component. Robitaille and Gauvin [34] proposed using two empirical power models for the compression and relaxation curves, to characterize material behavior. In compression, the fiber volume content depends on the effective stress $\sigma_{f}$ acting on the fiber bed as follows:(1)$$V_{f}=A\cdot \sigma_{f}^{B}$$
where $V_{f}$ is the fiber volume fraction, A is the fiber volume fraction at $\sigma_{f}=1\;Pa$, and B is the stiffness index of the compaction curve. The relaxation curve can be described as follows:(2)$$\frac{\sigma_{f}}{\sigma_{f0}}=1-C\cdot {t_{r}}^{\left(1/D\right)}$$
where $t_{r}$ is the relaxation time, $\sigma_{f0}$ is the level of stress applied initially before relaxation, C is the stress decay at $t_{r}=1s$, and D is the relaxation index.

Our previous paper [35] discussed compression and relaxation tests with glass fiber stacks; the compaction load at the same target thickness (3 mm) was similar in both states 392.8 ± 38.7 kPa in dry conditions and 359.6 ± 40.9 kPa in saturated conditions. The results indicated that the less pressure was needed for wet preform than dry preform to reach the same thickness. When the specimens were saturated with resin, the fabric preform would normally undergo reduction in average thickness at the same pressure due to the lubrication effect; this conclusion has already been verified by many researches [36,37].

### 2.2. Hydromechanical Coupling During Infusion

The resin flow through a porous medium such as fabric preform is governed by Darcy’s law:(3)$$v=-(\frac{K}{\eta })\cdot \nabla P$$
where v is the resin volume flow rate, K is the permeability tensor of the fabric preform, η is the dynamic viscosity of the resin, and ∇P is the driven pressure gradient before and after the flow front. For the in-plane unidirectional flow, the relation between the flow front and the injection time along the x axis is given by:(4)$$t=\frac{\eta (1-V_{f})}{2K_{x}∆p}{x(t)}^{2}$$
where t is the injection time, $K_{x}$ is the permeability of the fabric preform along the x axis, ∆p is the pore pressure, and x(t) is the position of the flow front.

During the resin impregnation stage of the VARI process, the advance of the resin flow front is an important feature. According to Terzaghi’s law [38], the atmospheric pressure ($P_{a}$) applied by the vacuum system is determined by the effective stress on the fiber bed ($\sigma_{f}$) and the pore pressure (p):(5)$$P_{a}=\sigma_{f}+p$$
which means that when the resin arrives, the liquid pressure increases and the load $\sigma_{f}$ on the preform decreases.

### 2.3. Weak FBGs Array Monitoring Principle

Compared with conventional FBGs, the weak FBG array has the advantages of high sensitivity and more sensing points due to the sharp spectrum and narrow band- width [39] The inline fabrication technology using a draw tower ensures the fiber Bragg gratings are constructed at the same time as fiber preparation (before fiber coating), in order to ensure the integrity of the OF structure and avoid the decrease in mechanical strength and loss of light intensity caused by the optical fiber pretreatment and welding of multiple gratings.

For a single FBG sensing point, the wavelength (λ) is calculated by the period of the grating (Λ) and the effective refractive index of the core of OF ($n_{eff}$):(6)$$\lambda =2n_{eff}\Lambda$$

We carried out the temperature monitoring test during the resin-filling stage in the VARI process, at room temperature. Three thermocouples were embedded in different layers inside the GFRP sample. The temperature inside the GFRP changed less than 0.4 °C during the whole resin-filling stage. So, it is reasonable to neglect the temperature effect during resin flow monitoring using an FBG sensor in the scenario in this manuscript. The local stress relaxation and small change in strain because of the arrival of the resin can be sensed by the FBG based on the stretching of the grating period and the elastic–optical effect:(7)$$∆\lambda /\lambda_{0}=K_{\varepsilon }\varepsilon_{axial}$$
where $\lambda_{0}$ is the initial wavelength of the grating and ∆λ is the wavelength change during the resin-filling stage. $K_{\varepsilon }$ is the sensitivity of axial strain $\varepsilon_{axial}$ depending on the material properties of the OF and can be obtained by the calibration test (1.1 pm/με in this paper). As shown in Figure 1, each FBG can be regarded as a flow sensor to monitor the local status of resin impregnation, and the FBG array can achieve quasi-distributed flow monitoring in real time.

## 3. The In-Plane Flow Monitoring with the OF Parallel to the Flow Direction

To investigate sensitivity and the ability to monitor the resin flow front, three kinds of fiber optic sensors were employed in this part of the study, including a polyimide-coated OF (CH1), a polyacrylate-coated OF (CH2), and a polyacrylate-coated OF with the coat stripped at the grating areas (CH3). Incorporating dense wavelength division multiplexing technology, each OF consisted of ten FBGs with a grating length of 10 mm and grating distance of 50 mm. Each OF could be used as ten flow sensors, with a central wavelength range of 1500–1600 nm for polyimide-coated OFs and 1529–1569 nm for polyacrylate-coated OF.

The vacuum-assisted resin infusion (VARI) process was used. Flow-front inspections were carried out with two cameras and one transparent mold. An aluminum frame was used to support the acrylic mold after it was aligned using a horizontal aligning level. Then, twelve layers of woven glass fabric manufactured by Weihai Guangwei Composites Co., Ltd. (No. 307, Wenhua West Road, Torch Hi-tech Industry Development Zone, Weihai, China) were cut with a fabric shear to the size 300 mm × 200 mm and stacked. From bottom (mold contacted with 0 layer) to top (vacuum bag contacted with 12 layer), three OFs were looped, then pasted on the mold (FBG 1, 3, 5, 7, 9 of CH1 and FBG 1, 2, 3, 4, 5 of CH2 and CH3) and embedded into layer 5 to 6 (FBG 10, 8, 6, 4, 2 of CH1 and FBG 6, 7, 8, 9, 10 of CH2 and CH3) of the fabric stack; see Figure 2. The numbers in circles in Figure 2b indicate the numbers of FBGs for each OF. The distance between two OFs was 30 mm, which was more than 100 times of the diameter of each OF and large enough to avoid the influence of neighboring OFs. Two ends of the OF were pasted with tape to straighten the sensors and position them parallel to the weft. Then, the release film and flow medium were placed at the top of the fabric stack. The injection line, breather, and vent were also placed at preset locations.

Then, the system was vacuumed with a single vacuum bag and the OF was connected to the TV125 interrogator with a resolution of 1 pm and wavelength range of 1500~1600 nm (manufactured by Tongwei Technology Co., Ltd., Beijing, China). One issue that was addressed is that the vacuum bag had to be kept flat in the FBG-monitored areas to avoid small flow channels induced by wrinkles and to maintain the uniform stress applied on the gratings. To avoid curing the resin, only EC-TDS-IN2 Infusion Resin provided by EasyComposites without the curing agent was infused via a tube of inner diameter 4 mm. Two cameras were used at the top and bottom to monitor the flow fronts, and the spectral signals of OF were collected simultaneously at a frequency of 2 Hz. The schematic diagram of the experimental setup, the sensor layout, and the top and bottom views of the flow fronts are shown in Figure 2.

The spectral signals of OF of CH1 and CH3 before and after resin injection are shown in Figure 3. While the polyimide-coated weak FBGs were prepared in-line fabrication technology using a draw tower, the adjacent gratings were made in a spaced wavelength to avoid the mutual interference. For example, the first FBG of the polyimide-coated OF was actually the grating 2 with the central wavelength of 1542 nm, see Figure 3a. The polyacrylate-coated OF was in the serial grating numbers from 1 to 10, see Figure 3b. We observed that the response signals of each FBG remained undistorted. The central wavelength of each FBG showed a small shift due to the slight effect of resin flow and pressure release, which did not disturb other FBGs and was easy to extract. The find-peaks function in MATLAB was used and the central wavelengths of each FBG at different injection times were extracted.

Therefore, the strain curve of each FBG was calculated by Equation (7), and these are shown in Figure 4, except FBG10 of CH1, which was right in the edge of the sample and the signal was heavily disturbed. By comparing with the results monitored by the top and bottom cameras, the singular points of the strain curves were extracted and analyzed. Two kinds of trends of strain curves were found according to the placement of the OFs. For the middle FBGs surrounded by deformable and dry fabric, the resin was quite far away from the gratings, and the OFs were impregnated as a whole. The wet fabric was easier to compact than the dry fabric; therefore, the OFs were better bonded and the preform was compressed harder, which resulted in a decrease in strain in the early stages. When the resin flowed near the grating area, the vacuum pressure released partially to actuate the resin; so, the stress on the fiber bed decreased. Therefore, the two mechanisms including the harder compaction of wet fabric and stress relaxation were in competition with each other, resulting in a complex strain curve. When the resin arrived, the factor of stress decrease dominated and the strain began to increase. This was also the reason that the local preform thickness increased when the resin became involved in the VARI process. For the bottom FBGs in contact with the nondeformable mold, the strain remained unchanged before the resin arrived. The onset points where the slope began to increase sharply were defined as the times when the resin arrived, as shown in Figure 4a–f. Different resin arrival times were obtained for each FBG and the curves clearly showed the sequences.

With the advancement of the resin and movement of the flow front away from the injection line, the pressure gradients before and after the flow front become smaller and smaller [37]. As shown in Figure 4d,f, the changes in strain at different FBGs showed the trend FBG6 > FBG7 > FBG8 > FBG9 > FBG10, which also corresponded to the pressure gradients. For FBGs at the same x coordinates (the distance from the injection line), the changes in strain due to the resin flow were compared; the first group included CH1-FBG7 (20.1 με), CH2-FBG4 (34.2 με), CH3-FBG4 (37.7 με) and the second group included CH1-FBG8 (166.4 με), CH2-FBG7 (175.4 με), CH3-FBG7 (157.3 με). We can conclude that the polyacrylate-coated OF was a little more sensitive to the resin flow than the polyimide-coated OF, due to the weaker bond and easier water absorption, and changed strains of polyacrylate-coated OF were slightly higher than the polyacrylate-coated OF with the coat stripped at grating areas.

To validate the monitoring ability of the FBG array, the flow fronts monitored by the bottom camera were extracted and compared with the FBG-monitored results, as shown in Figure 5a–c. The good compatibilities can be found between the FBG array and bottom-camera-monitored results. The results show that the change onset points of the strain curves can be used to track the resin flow fronts.

## 4. The In-Plane Flow Monitoring with the OF Perpendicular to the Flow Direction

In the real manufacturing processes of composite structures, the double-vacuum-bag method is sometimes used to avoid air leakage and to achieve higher fiber volume fraction. In this section, the in plane flow front in a VARI process with double vacuum bags was monitored using the FBG array perpendicular to the flow direction. In order to minimize the influence of the embedded OFs on resin flow and the mechanical properties of the cured composite material, smaller-sized polyimide-coated OFs were used in the rest of tests, with the compromise of less sensitivity.

Twelve layers of woven glass were stacked with the dimensions of 800 mm × 250 mm. There were 15 OF with four FBGs numbered FBG-1,3,5,7 and each OF was embedded into the 6 and 7 layer of the fabric stack. The OF were straightened and placed carefully paralleled to the weft with the help of the forming spray provided by Minnesota Mining and Manufacturing Co. In order to achieve the subsequent health monitoring of the composite structure during service, each OF was protected and encapsulated with capillary at the in and out places ingress and egress of reinforcement. The capillaries were all sealed at two ends using the vacuum seal grease to avoid resin leakage into the tube and reducing the toughness of OF. Unlike the sensitivity test, the system was sealed with the double-vacuum-bag method, which meant that the top vacuum bag was always compacted by atmospheric pressure. Then, the capillaries were sandwiched between two layers of sealant tapes. After the system was vacuumed, the leak test was conducted with less pressure change than 1.5 kPa up to 15 min. Until the leak tests were passed and then, the resin filling continued. The flow direction was perpendicular to the OFs and the inner diameter of the injection tube was 20 mm. Regarding the FBG sensors layout, there were two reasons 1. to minimize the channel-flow effect induced by the embedded optical fiber, because the resin can flow easier along the optical fiber if the resin flow paralleled to the sensor; 2. to validate the resin flow monitoring ability by comparison with different FBG sensors with one optical fiber. Using this sensor location, the FBG sensors with one optical fiber had similar responses when the linear resin flow arrived, which was further verified in the tests with linear injection. The detailed experimental setup and scene illustration can be found in Figure 6a,b.

By the same data processing method, the changed strain curves of the FBGs array were obtained and are shown in Figure 7. The third trend was found and differed from the results for VARI using a single vacuum bag. Before injection, the strain remained unchanged and the time when the injection opened was set to be zero. The OF was perpendicular to the flow direction, in order that the OF would not be partial impregnated. Therefore, when the resin was far away from the gratings (injection time 0–100 s of CH10), the strain still remained stable; see Figure 7a. Only when the resin arrived were the four FBGs of one OF all impregnated, and the strain curves of the same OF changed at the same time. The compaction pressure provided by the top vacuum bag was constant and the OF was compressed harder due to the fact that the wet woven fabric was easier to compress than dry fabric at the same pressure level. That was the reason the strain began to decrease, and the change onset indicated the time when the resin arrived. In Figure 7a, the flow front arrival times indicated by FBGs with one single OF showed very good consistency. With the FBG array of different OFs, flow-front monitoring at different positions was achieved, as shown in Figure 7b. At the injection time of 721 s, the resin-filling process finished and the vacuum pump was closed. Different strain curves slightly increased due to the relaxation of the wet fabric stack.

Due to the presence of the flow medium on the top of the fabric stack, the top flow was faster than the middle or lower flows. The results monitored by the top and bottom cameras were plotted and compared with the FBG array-monitored results; see Figure 8. The results met the expectations that the resin arrived times in the middle layer were between that in the top and bottom layers, which indicated the flow monitoring ability of FBG array for the VARI process with the double vacuum bag method. To conclude this section, different strain curves may be obtained during the VARI process, and appropriate resin flow-front monitoring criteria should be chosen based on the different placement setups of embedded OFs, resin flow direction, and the different vacuum bagging methods.

## 5. Out-of-Plane Flow Monitoring

### 5.1. Experimental Setup of Out-of-Plane Flow Monitoring

After the in-plane flow-front monitoring was achieved, the out-of-plane flow-front monitoring test was carried out. As shown in Figure 9a,b, three OFs including 30 FBGs were embedded into a50 layers glass fabric stack of size 250 mm×250 mm×8.2 mm. The OFs were in the middle of the x direction and in a different layer in the thickness direction. An acrylic mold with a central injection hole of 8 mm in diameter was employed and the resin was injected into the preform by a feed line with an inner diameter of 4 mm. The breather was placed on the left side and in partial contact with the preform to provide an elimination channel for the air inside. Then, the system was vacuumed with a single vacuum bag. The spectral signals of different OFs were collected and the top camera was used to visually monitor the flow states of the upper surface.

### 5.2. Experimental Results of Out-of-Plane Flow Monitoring

After the data processing, the strain curves of different FBGs were obtained and are shown in Figure 10. Because the FBG array was entirely surrounded by the soft glass fiber and a single vacuum bag method was used, most strain curves showed the same trends as the curves in Figure 4b,d, decreasing first due to the harder compaction of the wet preform and then increasing due to the stress relaxation. Therefore, the flow-front monitoring criteria were also the same as in Figure 4b,d. As shown in Figure 10, the resin arrival times could be obtained for most of the strain curves based on the change onset, except CH3-FBG2 and CH3-FBG6, which showed two descent stages. The first descent stages of CH3-FBG2 and CH3-FBG6 were nearly coincident with the descent stage of CH3-FBG5 and CH3-FBG4; so, a reasonable inference is that the first descent stages of CH3-FBG2 and CH3-FBG6 were caused by the partial release of stress while the resin was injected into the central of preform. The second descent stages were bigger and revealed the arrival of the resin. The monitoring results from central FBGs are collected in Figure 10d; the minimum strains showed a gradual decreasing trend except CH1-FBG4 in layer 20, which corresponded to the gradual release of stress caused by resin flow in the thickness direction.

The in situ monitoring results of the out-of-plane flow fronts are summarized in Figure 11. The *x* axis indicates the distance from the center of injection, the *y* axis the layer location where the sensor was placed, and the *z* axis is the resin arrival time. On the YZ plane projection surface, the same trends of quadratic curves were found for results of sensors at the same distance from the injection center. It was not possible to compare the internal FBG-monitored results with those from the camera; so, only the upper FBGs were compared and are listed in Table 2. The errors of CH3-FBG4,2,6 were 4%, 17%, and 7%, respectively. In situ out-of-plane flow monitoring was achieved by the weak FBG array.

## 6. Defects Monitoring

### 6.1. Experimental Setup of Defects Monitoring

In this section, the ability of the FBG array to detect manufacturing defects including dry spots and voids was tested. The defects were introduced by cutting the flow medium into a predefined shape and creating areas of slow flow [40]. Unlike the carbon fabric with low permeability used in the Ref. [40] glass fabric was an order of magnitude higherpermeability, so a bigger notch in the flow medium was made with a size of 180 mm×100 mm, for the preform of 300 mm×210 mm×12 Layers. Three polyimide-coated OFs were looped and embedded into layer 8 to 9 (FBG 1, 3, 5, 7, 9) and layers 3 to 4 (FBG 10, 8, 6, 4, 2) of the fabric stack; see Figure 12. The feed line was placed on the left side and the system was vacuumed using one single vacuum bag. The captured flow fronts shown in Figure 12a indicated that an area of slow flow was created, which provided a good chance for us to analyze the detailed relations between the flow-front positions and the strain curve. The differences in flow rate with/without flow medium also introduced dry spots and trapped air successfully, and voids were located exactly at CH1-FBG1 and CH1-FBG2; see Figure 12b. The monitoring results for dry spots and trapped air are analyzed in the following part.

### 6.2. Experimental Results of Defects Monitoring

The monitoring results for FBG8 across three channels and the detailed flow fronts at different injection times are plotted in Figure 13. The bright and dark areas indicate the macroscopic-scale impregnated and non-impregnated areas, respectively. Some commonalities of the strain curves were found: (a) the resin arrival times corresponded to the bottom of the strain curve with a “V” shape; (b) as soon as the resin made contact with the grating, the strain began to increase; (c) the strains kept increasing even the resin flow pass or even away from the grating areas. In the camera results, only the macroscopic-scale flow could be observed, and the strain curves continuously increased because the microflow still existed and the stress kept being released. In the CH3-FBG8-monitored area with flow medium on the top, the resin flowed fast and passed the grating area in about 15 s. For CH2-FBG8 and CH1-FBG8 in the area of slow flow, 31 s and 84 s were needed to flow pass the gratings. The flow rates were positively correlated with the change rates of the strain curves and negatively correlated with the opening length of the strain curves with a “V” shape. No significant disturbance was observed at the flow front due to the flow channel introduced by the OF, which indicated that the OF that was smaller than 150 μm in diameter had almost no effect on the macroscopic resin flow.

As can be observed in Figure 12b, a dry spot was located near CH1-FBG4 and CH1-FBG6 and the trapped air was located exactly at CH1-FBG1 and CH1-FBG2. For the dry spot, the results from the adjacent CH1-FBG3,4,5,6 are plotted in Figure 14a. The maximum values of the changes in strain of CH1-FBG3 and 5 were 109.6 με and 63.9 με. For CH1-FBG4 and 6, only 34.7 με and 31.2 με changes in strains were observed. It was observed that the closer the FBGs were to the dry spot, the smaller stress relaxation and change in strain were induced by the poor state of impregnation. For CH1-FBG1, located at the small void, the change ind strain was 285.7 με, which was 20 με smaller than CH2-FBG1, under the same conditions in terms of the layer placement, position, and distance from the cut flow medium. For CH1-FBG2, located at the larger void, the change in strain was 47.3 με, which was about half the change in strain for CH2-FBG2. When the void/dry spot formed at an FBG sensor location, the less the pressure changed compared with fully impregnated areas, which also resulted in less change in strain. Therefore, the changed strain due to the arrival of the resin flow front can be used as an index for defects, as verified by the final image in the resin-filling stage and also the strain curves. Therefore, the maximum changed strain can be used to represent manufacturing defects including dry spots and voids by comparison with FBGs under the same conditions.

## 7. Discussions and Conclusions

With the advantages of small size and little effect while embedding, OFs are also widely used in cure and health monitoring throughout the life cycles of composite structures. The mechanical properties of GFRP (glass fiber-reinforced polymer) with or without 150 μm polyimide-coated OFs embedded were tested. Three test standards including GB/T-1447-2005, ASTM D6641-2016 [41], and GB/T-1449-2005 were used to fabricate the tensile, compressive, and bending GFRP samples. The thickness of the bending sample was 2.0 mm, consisting of 10 layers of glass fabric, and the others’ thickness was 2.4 mm, made of 12 layers of glass fabric. The OFs were all placed in the middle layer and on the central line of the samples parallel to the length direction. For each kind of test, at least five samples were tested. The results are summarized in Table 3. The abbreviations “TS”, “TM”, “CS”, “CM”, “BS”, “BM”, and “VC” stand for “tensile strength”, “tensile modulus”, “compression strength”, “compression modulus”, “bending strength”, “bending modulus” and “variation coefficient”, respectively. Higher variation was introduced with the embedded OF, and the VC of samples with OF was higher than the samples without OF. The ratios of change in mechanical properties were less than ±5%, except for the compression strength, which was acceptable for application.

In our work, the FBG array’s monitoring abilities for resin flow front and manufacturing defects in the VARI process were fully investigated. The monitoring criteria based on the change onset were determined from the camera monitoring results, as follows. (a) For FBGs placed directly on the rigid mold with a single vacuum bag: the point when the strain began to increase; (b) for FBGs placed between soft fabric with a single vacuum bag: the bottom of the “V”-shaped strain curve; (c) for FBGs placed between soft fabric with double vacuum bags: the point when the strain began to decrease. The results of the 2D and 3D flow monitoring tests matched well with the camera monitoring results. The changed strain could represent the manufacturing defects by comparing with the FBGs in the same conditions. Therefore, the weak FBGs array-based method is an effective way to achieve the quasi-distributed monitoring of resin flow and manufacturing defects with high sensing density.

For a general case, the algorithm to obtain the flow front points is summarized as following: 1. Determine the location of the sensor and the vacuum bagging method; 2. Choose the right monitoring criterions; 3. Analysis the strain curves to obtain the flow front points. Due to the variability and complexity of strain curves, the analysis is mainly on expert experience now and we hope to make the code to do this analysis in the future.

The effect on resin flow of channel introduced by embedded OF needs to be evaluated and the compensation algorithm is a good choice for quantitative flow prediction in the manufacturing process. The method also showed the potential to experimentally characterize the local permeabilities and impregnated states under the fact that the non-uniformity of the fabric induced by the unavoided fibre distortions and nesting while stacking. This method also showed the potential to experimentally characterize local permeabilities and impregnated states, based on the non-uniformity of the fabric induced by unavoidable fiber distortions and nesting while stacking.

## Figures and Tables

**Figure 1 sensors-24-07637-f001:**
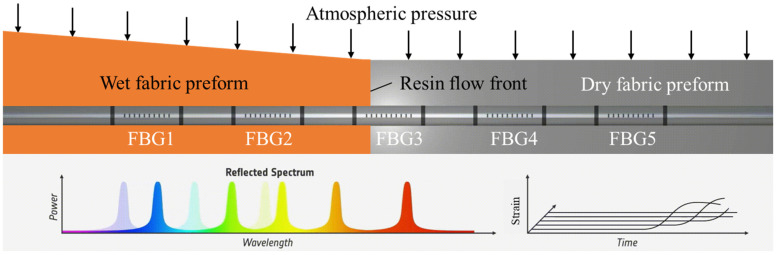
The flow-front monitoring principle of the FBG array during the VARI process.

**Figure 2 sensors-24-07637-f002:**
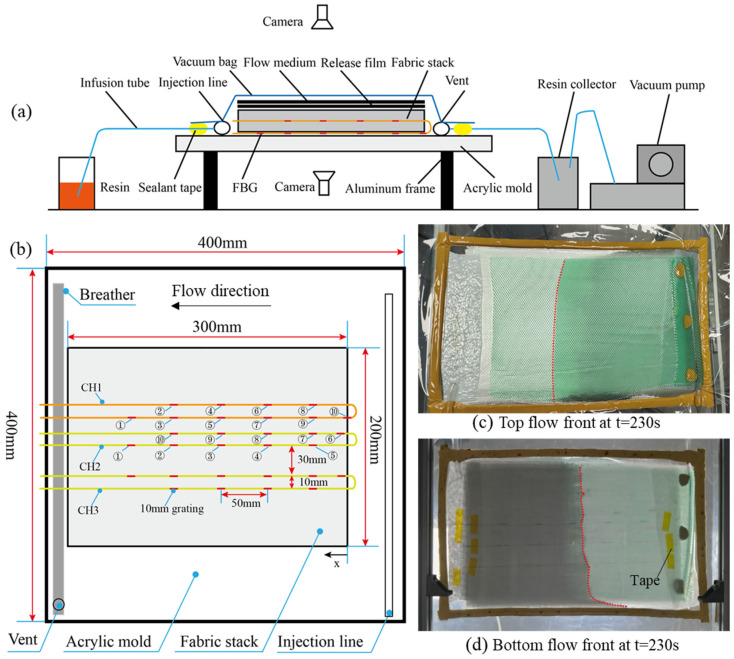
(**a**) The schematic diagram of the experimental setup, (**b**) the sensor layout, (**c**,**d**) the top and bottom view of the flow fronts.

**Figure 3 sensors-24-07637-f003:**
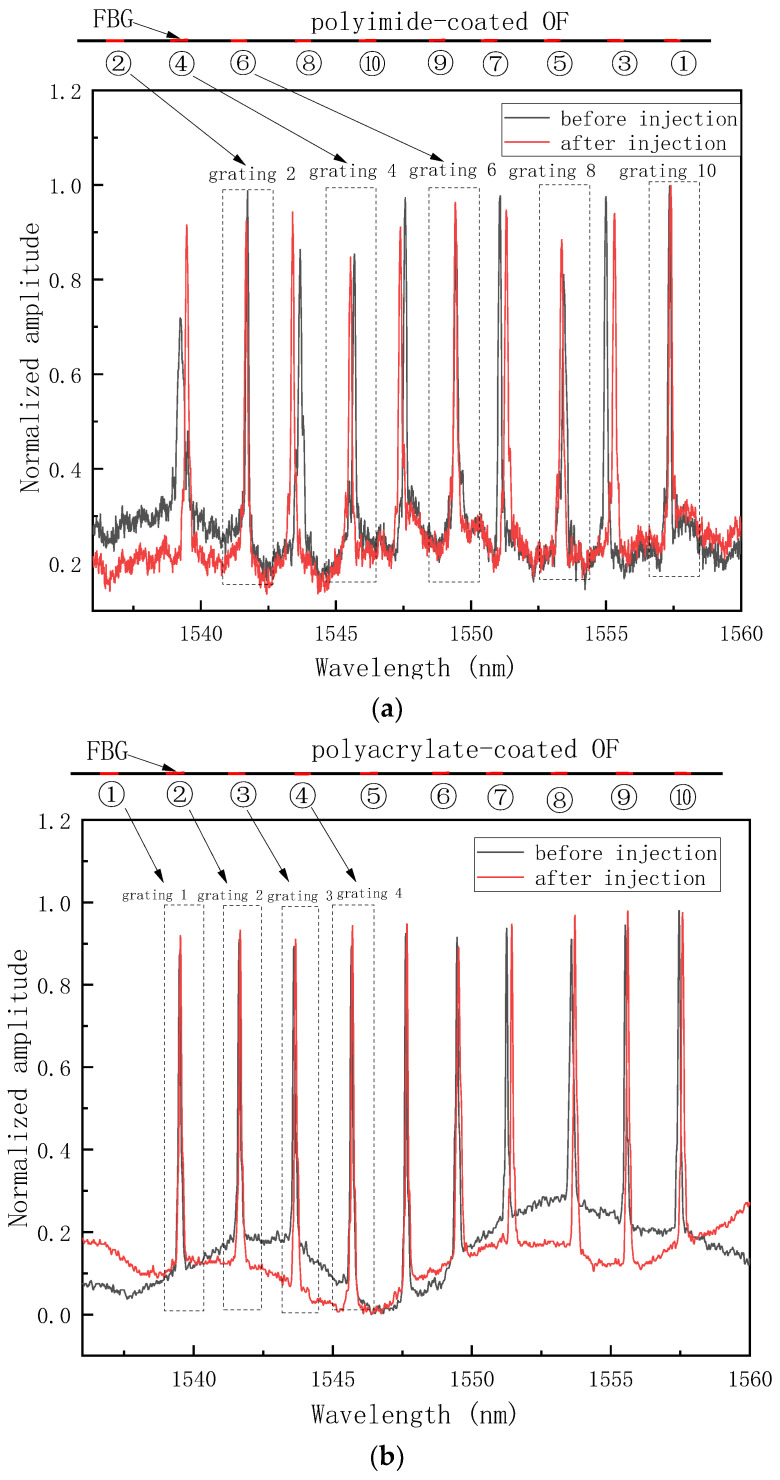
The spectral signals of different channels before and after injection. (**a**) CH1, (**b**) CH2, and (**c**) CH3.

**Figure 4 sensors-24-07637-f004:**
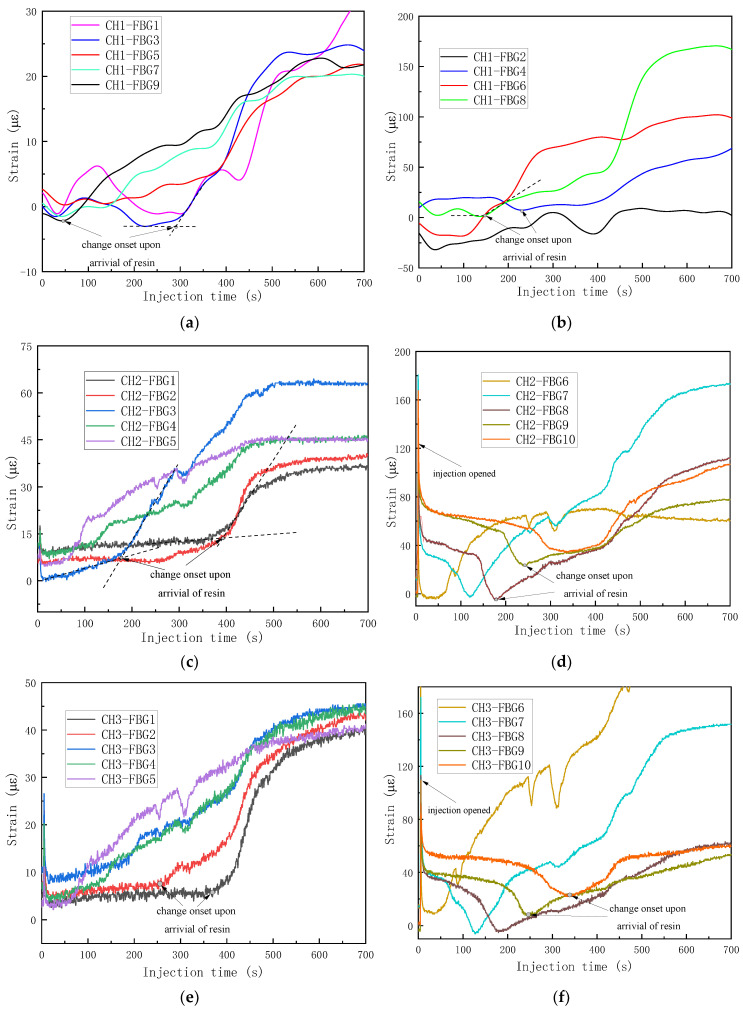
The strain curves obtained by the FBG sensor network during injection: (**a**) CH1-bottom FBGs; (**b**) CH1-middle FBGs; (**c**) CH2-bottom FBGs; (**d**) CH2-middle FBGs; (**e**) CH3-bottom FBGs; (**f**) CH3-middle FBGs.

**Figure 5 sensors-24-07637-f005:**
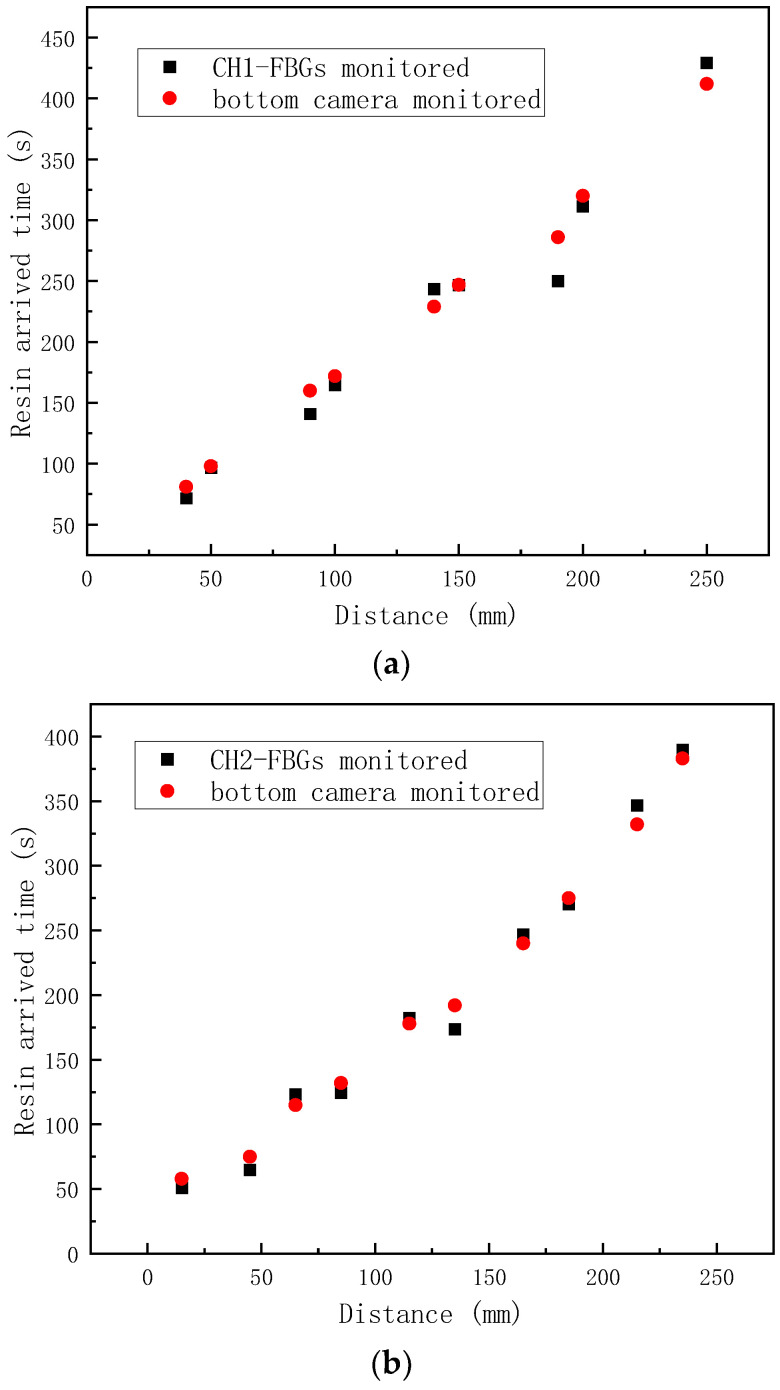
The in-plane flow fronts of VARI using a single vacuum bag, monitored by FBG array: (**a**) CH1-FBGs; (**b**) CH2-FBGs; (**c**) CH3-FBGs.

**Figure 6 sensors-24-07637-f006:**
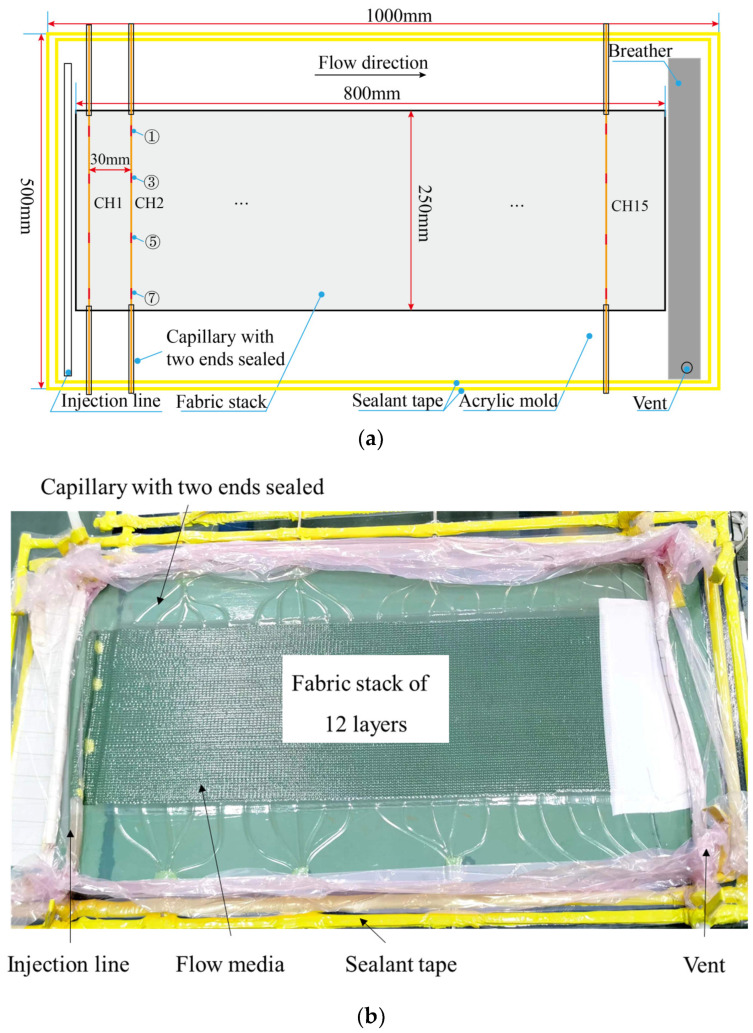
The experimental setup of the flow-front monitoring of the VARI process with double vacuum bags: (**a**) schematic diagram and sensor layout; (**b**) scene illustration.

**Figure 7 sensors-24-07637-f007:**
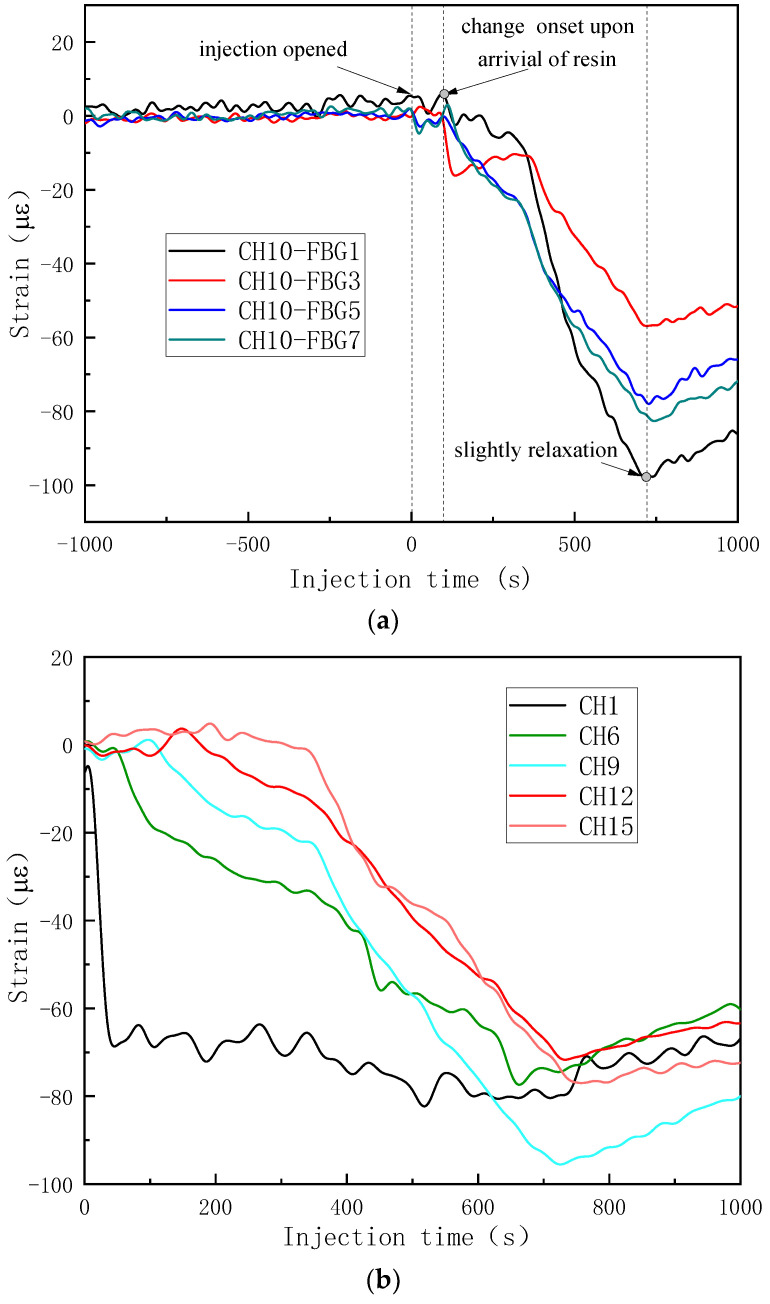
The changed strain curves obtained by the FBG array in the VARI process with double vacuum bags: (**a**) strain curves of same OF; (**b**) strain curves of different OFs.

**Figure 8 sensors-24-07637-f008:**
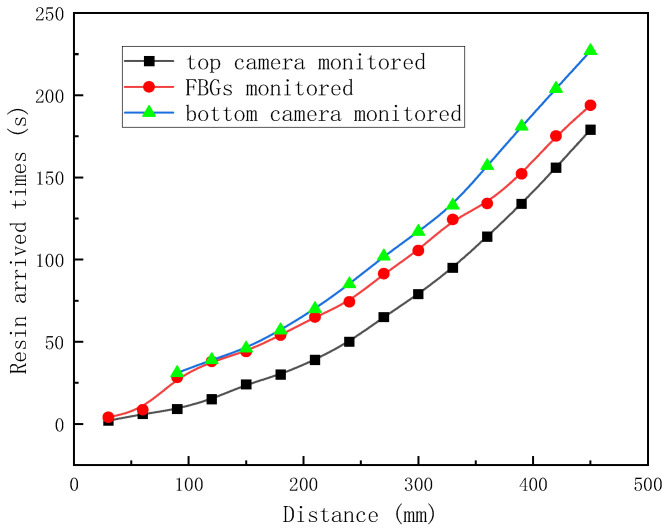
In-plane flow fronts of VARI using double vacuum bag, monitored by FBG array.

**Figure 9 sensors-24-07637-f009:**
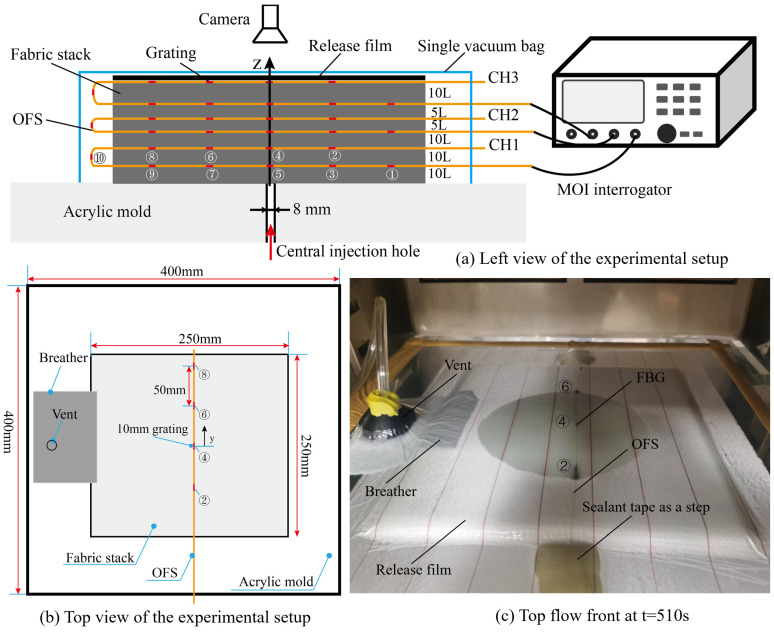
The experimental setup of out-of-plane flow-front monitoring of the VARI process with a single vacuum bag.

**Figure 10 sensors-24-07637-f010:**
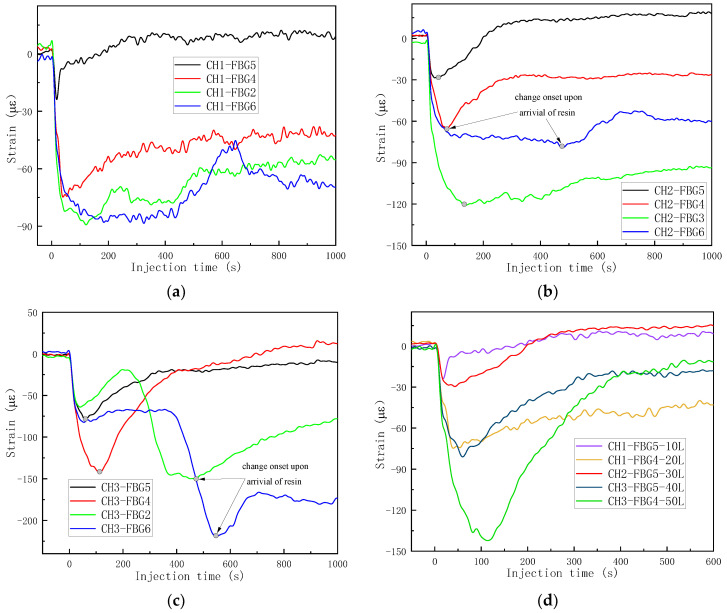
The strain curves in the out-of-plane flow monitoring experiment. (**a**) CH1; (**b**) CH2; (**c**) CH3; (**d**) central FBGs right up the injection.

**Figure 11 sensors-24-07637-f011:**
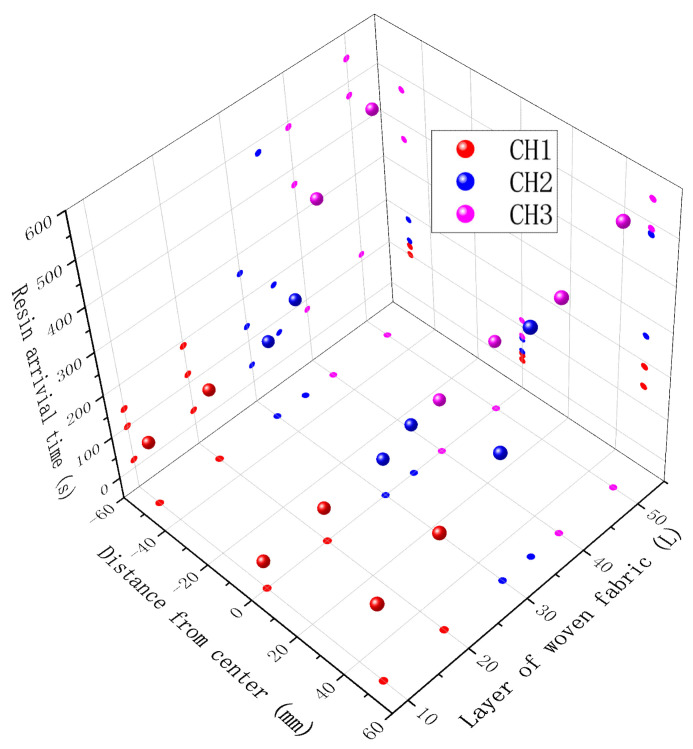
The in situ monitoring results of out-of-plane flow fronts during the VARI process.

**Figure 12 sensors-24-07637-f012:**
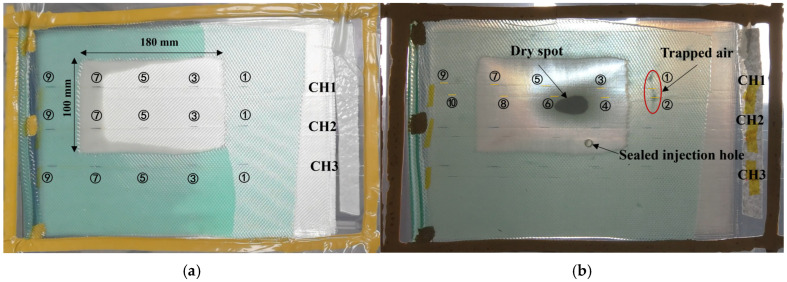
The captured flow fronts at different injection times in the defects monitoring test: (**a**) top view at t = 173 s; (**b**) bottom view at t = 700 s.

**Figure 13 sensors-24-07637-f013:**
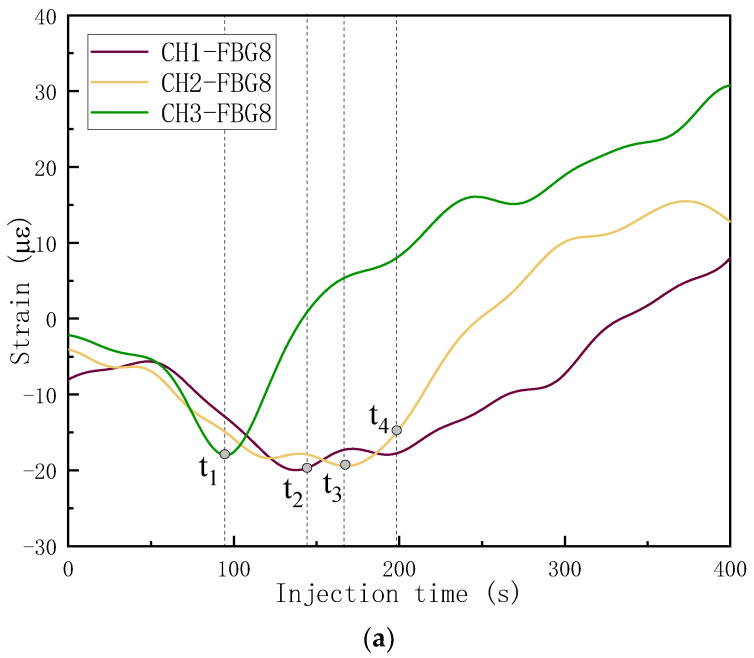
(**a**) The monitoring results; (**b**–**e**) flow front images at different injection times. (**b**) t1 = 95 s; (**c**) t2 = 145 s; (**d**) t3 = 167 s; (**e**) t4 = 198 s.

**Figure 14 sensors-24-07637-f014:**
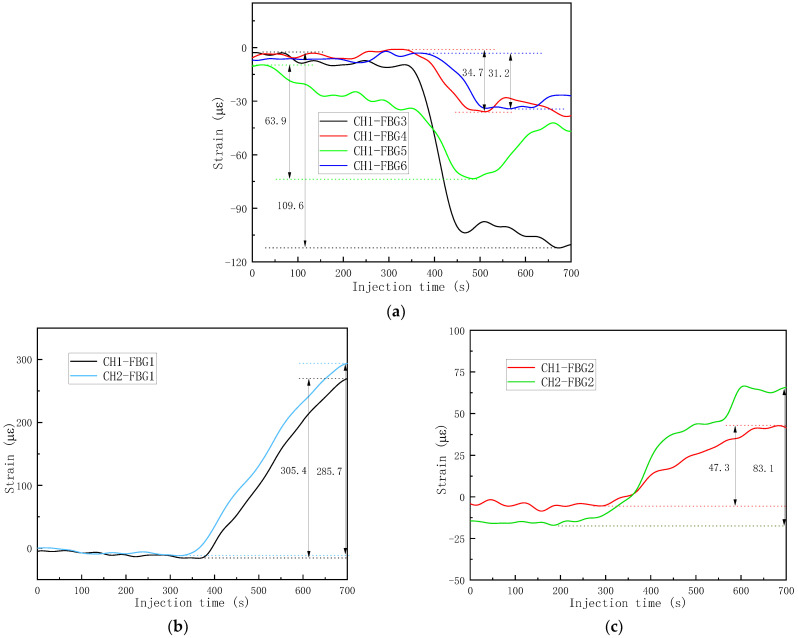
Representations of dry spot and void defects based on the FBG array: (**a**) FBGs near dry spot; (**b**) FBGs near small void; (**c**) FBGs near large void.

**Table 1 sensors-24-07637-t001:** Different in situ monitoring techniques for resin flow front.

Category	Measurement Method	Advantages	Disadvantages
Non-invasive	Camera [7,8]	Simple, low cost, and has no effect on the samples.	Needs transparent molds and the flow status inside cannot be obtained.
	Ultrasonic longitudinal wave [9,10]	Non-contact, suitable for all materials.	Limited accuracy and not suitable for sandwich composites.
Invasive	Pressure sensor [13]	Simple, low cost.	Large monitoring range required.
	Thermal sensor [14]	Low cost, durable.	Low accuracy and need for high-specific-heat resin.
	Electrical sensor [15,16,17,18,19]	High degree of integration.	Fewer measurement points, dependent on fiber types.
	conventional FBG sensor [24,25,26]	Small size and suitable for different kinds of fiber.	Fragile, costly, prone to signal loss, and limited sensing points.
	Distributed optical fiber sensor [32]	Small size and distributed monitoring ability.	Expensive and complex sensor and demodulator, short sensing length.
	Weak FBG array	Little effect on the host material, quasi-distributed monitoring ability.	Difficult to cut the finished product without breaking OF.

**Table 2 sensors-24-07637-t002:** The in situ monitoring results of out-of-plane flow fronts from the the upper FBGs.

Top FBGs	FBG Monitored Time	Camera Monitored Time	Error
CH3-FBG4	115.1 s	120 s	4%
CH3-FBG2	472.1 s	390 s	17%
CH3-FBG6	548.8 s	510 s	7%

**Table 3 sensors-24-07637-t003:** Comparison of the mechanical properties of GFRP with or without OF embedded.

Mechanical Properties	TS /MPa	VC /%	TM/GPa	VC /%	CS /MPa	VC /%	CM /GPa	VC /%	BS /MPa	VC /%	BM /GPa	VC /%
Without OF	642.5	1.93	26.12	1.59	334.8	3.6	28.29	4.5	395.26	1.82	20.07	1.63
With OF	622.7	3.77	25.76	3.12	297.2	8.8	27.83	9.1	409.12	2.52	21.10	2.90
Properties changed	−3.08%	----	−1.38%	----	−11.23%	----	−1.63%	----	3.51%	----	5.13%	----

## Data Availability

The data are available from the corresponding author on reasonable request.

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
