# Peer review of "Weak Fiber Bragg Grating Array-Based In Situ Flow and Defects Monitoring During the Vacuum-Assisted Resin Infusion Process"

_sensors, 2024, doi:10.3390/s24237637_

Round 1

Reviewer 1 Report

Comments and Suggestions for Authors

The weak FBG array was employed to monitor the real-time flow front and manufacturing defects during the VARI process. Detailed strain response when the resin arrived FBG sensors at different experimental setups such as the placement of OFS and the vacuum bagging methods were analyzed. The strain curves can provide effective information about the resin flow front and manufacturing defects. Therefore, the research is meaningful for the defects control in the real composite molding process. However, the follow questions should be properly addressed before acceptance of this work:

1.        Will the temperature during resin filling stage affect the strain monitoring results? Because the temperature and strain effects are strong coupling with each other for FBG sensor and the temperature must be compensated.

2.        The FBGs are oriented perpendicular to the direction of the resin flow. How to determine the location of the optical fiber, and what is the basis for such an arrangement?

3.        Does the capillary lead to the air leakage during VARI process? This is pretty tricky in the real manufacturing process.

4.        The explanation of the strain data for the void formation was not clear. To the reviewer it seems the image data from the cameras was the driver for interpreting the strain data and from the strain data alone it would be difficult to explain the void formation.

5.        Could the weak FBG array be used for the cure monitoring in the future study? And what are the advantages compared with other sensors?

6.        There are some studies on the composite monitoring using distributed optical fiber sensor. Compared to the following works, pls elaborated on the novelty of yours.

·       Bao X, Huang C, Zeng X, et al. Simultaneous strain and temperature monitoring of the composite cure with a Brillouin-scattering-based distributed sensor[J]. Optical Engineering, 2002, 41, 1496-1501.

·       Ito Y, Minakuchi S, Mizutani T, et al. Cure monitoring of carbon-epoxy composites by optical fiber-based distributed strain-temperature sensing system[J]. Advanced Composite Materials, 2012, 21, 259-271.

·       Sanchez M D, Gresil M, Soutis C. Distributed internal strain measurement during composite manufacturing using optical fibre sensors[J]. Composites Science and Technology, 2015, 120, 49-57.

·       Tsai J, Dustin S J, Mansson J. Cure strain monitoring in composite laminates with distributed optical sensor[J]. Composites Part A, 2019, 125, 105503.

·       Baocun Fan, Changhao Chen, Qi Wu*, et al, Temperature and strain monitoring during thermoforming of thermoplastic composite laminates using optical frequency domain reflectometry [J], Smart Materials and Structures, 2024, 33, 035021.

Reviewer 2 Report

Comments and Suggestions for Authors The manuscript studies the method of flow front monitoring during vacuum assisted resin infusion using fiber Bragg gratings. An array of FBGs is interrogated, and the measured strain is analyzed to detect the flow front and dry spot defects. The flow front propagates parallel, perpendicular, and circularly with respect to the sensing fibers. In my opinion, the manuscript contains useful information for readers working in the field and I recommend this manuscript for publication with revision. The following issues should be addressed: - The authors tried to put all information and data that they obtained into the text. I would recommend keeping the most significant part. For example, the introduction takes tree pages. It is not a review paper. There is no need to make a full paragraph describing one method or one type of sensor. Table 1 is redundant for a regular paper. Sections 2.1 and 2.3 also look unnecessary. - The authors use the abbreviation “OFS” in the sense of “optical fiber”. “Fiber” and “sensor” should be discriminated.   - Usually the last paragraph of introduction is used to describe what is going to be investigated in the paper. - It is not clear why the weak FBGs have the advantages of high sensitivity and more sensing points compared with the conventional FBG. Strong FBGs written during fiber drawing would have similar characteristics. Weak FBGs can be written after fabrication. So, what do we need: weak FBGs or in-line fabrication? Can we observe the results presented in the paper with normal FBGs? - Table 2 is quite useless. Everything is already described in the text. - Line 238. What is "0 and 1 layer"? - Why the numbering of sensors is so irregular? For example, in Fig.2 CH1, the order of sensors on the fiber is 2,4,6,8,10,9,7,5,3,1. Could you renumber them consecutively? - The English should be improved. - Line 250. What is MOI? - Why there is no CH2 in Fig. 3? - Fig. 3. What is meant by “original” spectra? Is it before applying vacuum or after? How do the spectra change when vacuum is applied? Did you ever observe birefringence due to transverse pressure? - Fig. 4. Why are some sensors missing: 1,10 in CH1, 2,6 in CH2, 5,6 in CH3? - How did you find the flow front point from the measured strain curve? Did you do it manually, knowing where the front was? Could you describe the algorithm how to do it in the general case. Is it possible in principle to automate this analysis? How did you get points for Fig.5 (a) from Fig.4 (a,b)? - Eq. 4 does not have the linear term. Why do you fit the data in Fig.5 with a curve where the linear term prevails? - Subsection 3.2 could be a new Section 4. The main feature here is that the fibers are parallel to the flow front, not double bag. - How thick was 50 layers? - Line 430 and Table 3. There is no need in the third and fourth decimals in error numbers. - Fig. 11. Axes titles are poorly visible. - Section “Discussion and conclusions” is strange. No discussions. Poor conclusions. It should be rewritten.

Comments on the Quality of English Language

The English should be improved.

Author Response

For research article

Response to Reviewer 2 Comments

1. Summary

2. Questions for General Evaluation

Reviewer’s Evaluation

Response and Revisions

Does the introduction provide sufficient background and include all relevant references?

Can be improved

Is the research design appropriate?

Yes

Are the methods adequately described?

Yes

Are the results clearly presented?

Yes

Are the conclusions supported by the results?

Can be improved

3. Point-by-point response to Comments and Suggestions for Authors

The manuscript studies the method of flow front monitoring during vacuum assisted resin infusion using fiber Bragg gratings. An array of FBGs is interrogated, and the measured strain is analyzed to detect the flow front and dry spot defects. The flow front propagates parallel, perpendicular, and circularly with respect to the sensing fibers. In my opinion, the manuscript contains useful information for readers working in the field and I recommend this manuscript for publication with revision. The following issues should be addressed: -

Comments 1: The authors tried to put all information and data that they obtained into the text. I would recommend keeping the most significant part. For example, the introduction takes three pages. It is not a review paper. There is no need to make a full paragraph describing one method or one type of sensor. Table 1 is redundant for a regular paper. Sections 2.1 and 2.3 also look unnecessary.

Response 1: Thanks for your comment. The introduction part has been modified and only takes two pages now. The Table 1 is a comprehensive summary of the in-situ monitoring techniques for resin flow front, which can be the guide for the readers. The section 2.1 provides the reason why the strain curves showed three different trends because two mechanisms including the harder compaction of wet fabric and stress relaxation will compete with each other when the resin flow arrived. And the section 2.3 is the flow monitoring principle of the weak FBG array and also provides the method for experimental data processing. Therefore, we request to keep these sections.

Please see the introduction part in the revised manuscript marked in red.

Comments 2: The authors use the abbreviation “OFS” in the sense of “optical fiber”. “Fiber” and “sensor” should be discriminated.

Response 2: Thanks for your comment. We changed the abbreviation “OFS” into “OF” in the whole text to discriminate the fiber and sensor.

Please see the revised manuscript marked in red.

Comments 3: Usually the last paragraph of introduction is used to describe what is going to be investigated in the paper.

Response 3: Thanks for your comment. The sentences have been added at the last paragraph of introduction: The novelty of our paper is to give a detailed spectrum and strain responses by using weak FBGs array during the resin filling stage of VARI process. Different strain curves with different setups were discussed and the resin flow front and manufacturing de-fects were also monitored using the weak FBGs array.

Please see the lines 164-167 of the revised manuscript marked in red.

Comments 4: It is not clear why the weak FBGs have the advantages of high sensitivity and more sensing points compared with the conventional FBG. Strong FBGs written during fiber drawing would have similar characteristics. Weak FBGs can be written after fabrication. So, what do we need: weak FBGs or in-line fabrication? Can we observe the results presented in the paper with normal FBGs?

Response 4: Thanks for your comment. In-line fabrication of FBG usually uses the single pulse, so the strong FBGs are difficult to be written during fiber drawing. The weak FBGs is more sensitive to the external disturbance caused by the resin flow due to the sharp spectrum [39]. On the other side, the weak FBGs have narrow bandwidth and can realize thousands of sensing points through mixed wavelength division and time division multiplexing methods, which are more suitable for large-scale monitoring compared with strong FBGs. The same strain curves can be obtained by strong FBGs. However only limited sensing points can be got. That is why we used weak FBGs array in this paper and tried to achieve quasi-distributed flow front monitoring during the manufacturing process.

Please see the lines 213-214 of the revised manuscript marked in red.

Comments 5: Table 2 is quite useless. Everything is already described in the text.

Response 5: Thanks for your comment. The related text has been deleted because the Table 2 is clear and intuitive: Thanks to the Dense Wavelength Division Multiplexing technology, each OF consisted of ten FBGs were used and the properties of two kinds of OFs were listed in Table.2.

Please see the lines 245-249 of the revised manuscript marked in red.

Comments 6: Line 238. What is "0 and 1 layer"?

Response 6: Thanks for your comment. The FBGs embedded between 0 and 1 layer means that they are under the bottom layer and contact with the mold. The sentences are modified as following:

Three OFs were looped, then pasted on the mold (FBG 1,3,5,7,9 of CH1 and FBG 1,2,3,4,5 of CH2 and CH3) and embedded into 5 to 6 layer (FBG 10,8,6,4,2 of CH1 and FBG 6,7,8,9,10 of CH2 and CH3) of the fabric stack, see Fig. 2.

Please see the lines 256-259 of the revised manuscript marked in red.

Comments 7:Why the numbering of sensors is so irregular? For example, in Fig.2 CH1, the order of sensors on the fiber is 2,4,6,8,10,9,7,5,3,1. Could you renumber them consecutively?

Response 7: Thanks for your comment. While the polyimide-coated weak FBGs were prepared in-line fabrication technology using a draw tower, the gratings were made with the numbers of 2,4,6,8,10,9,7,5,3,1 to avoid the mutual interference. I am afraid the order of sensors cannot be changed.

Comments 8: Line 250. What is MOI?

Response 8: Thanks for your comment. The MOI is the short for interrogator. Now the interrogator scheme and the manufacturer have been added: Then vacuum the system with single vacuum bag and connected the OF to the TV125 interrogator with the resolution of 1 pm and wavelength range of 1500~1600 nm (manufactured by Tongwei Technology Co., Ltd).

Please see the lines 270-272 of the revised manuscript.

Comments 9: Why there is no CH2 in Fig. 3? - Fig. 3. What is meant by “original” spectra? Is it before applying vacuum or after? How do the spectra change when vacuum is applied? Did you ever observe birefringence due to transverse pressure? -

Response 9: Thanks for your comment. The spectral signals of CH2 have been added. The title of Fig.3 has been changed as following: The spectral signals of different channels before and after injection. (a) CH1,( b) CH2, and (c) CH3. The signals were obtained after applying vacuum and before injection. The central wavelength would shift left due to the compression force when the vacuum is applied. The pressure is quite small. The vacuum bag is soft and can be easily deformed. Therefore, the transverse pressure acted on the grating area is small and we never observe the birefringence during the infusion process. However, the birefringence appeared in the post-cure stage at the cooling stage from 100 ℃ to room temperature due to the big thermal residual stress.

Please see the Figure.3 of the revised manuscript.

Comments 10: Fig. 4. Why are some sensors missing: 1,10 in CH1, 2,6 in CH2, 5,6 in CH3? How did you find the flow front point from the measured strain curve? Did you do it manually, knowing where the front was? Could you describe the algorithm how to do it in the general case. Is it possible in principle to automate this analysis? How did you get points for Fig.5 (a) from Fig.4 (a,b)? -

Response 10: Thanks for your insightful comment. The results of sensors showed the same trends. For the better display effect , the results of FBGs 1,10 in CH1, 2,6 in CH2, 5,6 in CH3 were not showed in Fig.4. Then the flow front points were obtained by the strain curves directly from the strain curves. When the resin flowed near the grating area, the vacuum pressure released partially to actuate the resin, so the stress on the fiber bed decreased. Therefore, two mechanisms including the harder compaction of wet fabric and stress relaxation will compete with each other, resulting in a complex strain curve. When the resin arrived, the factor of stress decrease was dominating and the strain began to increase, and that is also the reason the local preform thickness increases when the resin comes in the VARI process. For the bottom FBGs contacted with the nondeformable mold, the strain remained un-changed before resin arrived. The onset points where the slope began to increase sharply were defined as the time when the resin arrived, as shown in Fig. 4. The flow points of Fig.4(a)-(f) have been added. Different resin arrival times could be obtained for each FBG and the curves clearly showed the sequences.  

For a general case, the algorithm to obtain the flow front points is summarized as following: 1. Determine the location of the sensor and the vacuum bagging method;

2. Choose the right monitoring criterions: the strain began to increase or decrease;

3. Analysis the strain curves to obtain the flow front points.

Due to the variability and complexity of strain curves, it is difficult to do the analysis automatedly. Now the analysis is mainly on expert experience and we hope will make the code to do this analysis automatedly in the future.

Please see the Figure.4 and lines 565-571 of the revised manuscript.

Comments 11: Eq. 4 does not have the linear term. Why do you fit the data in Fig.5 with a curve where the linear term prevails? -

Response 11: Thanks for your insightful comment. The Fig.5 has been modified with the quadratic fitting.

Please see the Figure.5 and lines 565-571 of the revised manuscript.

Comments 12: Subsection 3.2 could be a new Section 4. The main feature here is that the fibers are parallel to the flow front, not double bag. -

Response 12: Thanks for your comment. The Subsection 3.2 has been changed into a new Section 4 titled “The In-Plane Flow Monitoring with the OF Perpendicular to the Flow Direction”.

Please see the line 355 of the revised manuscript.

Comments 13: How thick was 50 layers? -

Response 13: Thanks for your comment. The thick of the cured sample with 50 layers is 8.2 mm. The sentence has been modified as following:

Three OFs including 30 FBGs were embedded into the 50 layers glass fabric stack with the size of .

Please see the lines 429-430 of the revised manuscript.

Comments 14: Line 430 and Table 3. There is no need in the third and fourth decimals in error numbers.

Response 14: Thanks for your comment. The errors of CH3-FBG4,2,6 were 4%, 17% and 7%, respectively.

Please see the line 470 and Table.3 of the revised manuscript.

Comments 15: Fig. 11. Axes titles are poorly visible. -

Response 15: Thanks for your comment. The Fig 11 has been rotated to make the axes title clearly visible.

Please see the Fig.11 of the revised manuscript.

Comments 16: Section “Discussion and conclusions” is strange. No discussions. Poor conclusions. It should be rewritten.

Response 16: Thanks for your comment. The Discussion part is the mechanical properties comparison of GFRP with or without OF embedded to see the practicability to achieve the structural health monitoring in the service stage with embedded FBG sensors. The algorithm to obtain the flow front points is summarized in conclusions part:

For a general case, the algorithm to obtain the flow front points is summarized as fol-lowing: 1. Determine the location of the sensor and the vacuum bagging method; 2. Choose the right monitoring criterions: the strain began to increase or decrease; 3. Analysis the strain curves to obtain the flow front points. Due to the variability and complexity of strain curves, it is difficult to do the analysis automatedly. Now the analysis is mainly on expert experience and we hope will make the code to do this analysis in the future.

Please see the lines 571-577 of the revised manuscript.

4. Response to Comments on the Quality of English Language

Point 1: The English should be improved.

Response 1: Thanks for your comment. We had gone through the whole text and modified some clerical and grammatical errors.

5. Additional clarifications

There are no additional clarifications here.

Reviewer 3 Report

Comments and Suggestions for Authors

The authors have provided an interesting and timely paper on fiber-optic sensors. What is important, this paper has practical significance. I recommend this manuscript for publication in Sensors after correction of some shortcomings:

1. I would ask the authors to expand the literature review to present a wider range of distributed and quasi-distributed sensors that can solve this problem.

2. In the table comparing various sensor systems, it is not very clear what is meant by "conventional fiber-optic sensors"? If it is optical frequency domain reflectometry (OFDR), then I find the mention of 'weak signal' questionable. Firstly, because the same FBGs can be interrogated using OFDR [http://dx.doi.org/10.1016/j.ijleo.2022.169764] and can give the same reflections, and secondly, there are schemes with erbium signal amplification [http://dx.doi.org/10.1134/S0020441223050172].

3. In the mentioned table, it is appropriate to make references to the literature, if any.

4. Since the experiment must be repeatable, I would ask the authors to provide the interrogator scheme or, in the case of using a commercial system, to indicate the manufacturer and brand of the instrument.

5. In Figure 1, the word 'perform' is written instead of the word 'preform', right?

6. Some figures and tables in the article are located right at the end of the section, I suggest raising them in the text above, immediately after the first mention.

Round 2

Reviewer 2 Report

Comments and Suggestions for Authors

Comments to the authors’ responses

Comment 1:  Some improvement is made in the introduction.  The authors insist to keep Table 1 and Sections 2.1, 2.3.

Comment 2: Ok.

Comment 3: Some improvement is made.

Comment 4: Ok. Although, the number of sensors described in the paper is 10, which can be done using normal FBGs. So, the advantage has not been used.

Comment 5: I would prefer to remove the table, which has just two almost identical rows. 

Comment 6: Not clear. How do you define “5 to 6 layer”?

Comment 7: Not clear. What does it mean “the gratings were made with the numbers”?

Comment 8: Ok.

Comment 9: Ok. Check line 282. What are the number at the Y axis in Figs. 3 b,c?

Comment 10: This is strange to omit some data without explanation. Every curve should have the point of resin arrival. In my opinion, it is not possible to find the flow front point from the strain curves measured by the authors if you do not know beforehand where the front is. Therefore, it is not clear how to monitor the flow front.

Comment 11: Do not see any changes.

Comment 12: Ok.

Comment 13: Ok.

Comment 14: Ok.

Comment 15: Ok.

Comment 16: No improvement.

4. Response to Comments on the Quality of English Language: The English still should be improved.

Comments on the Quality of English Language

The English should be improved.

Author Response

For research article

Response to Reviewer 1 Comments

1. Summary

2. Questions for General Evaluation

Reviewer’s Evaluation

Response and Revisions

Does the introduction provide sufficient background and include all relevant references?

Can be improved

Is the research design appropriate?

Yes

Are the methods adequately described?

Yes

Are the results clearly presented?

Yes

Are the conclusions supported by the results?

Can be improved

3. Point-by-point response to Comments and Suggestions for Authors

The manuscript studies the method of flow front monitoring during vacuum assisted resin infusion using fiber Bragg gratings. An array of FBGs is interrogated, and the measured strain is analyzed to detect the flow front and dry spot defects. The flow front propagates parallel, perpendicular, and circularly with respect to the sensing fibers. In my opinion, the manuscript contains useful information for readers working in the field and I recommend this manuscript for publication with revision. The following issues should be addressed: -

Comments 1: Some improvement is made in the introduction. The authors insist to keep Table 1 and Sections 2.1, 2.3.

Response 1: Thanks for your comment to improve the quality of the manuscript.

Comments 2: Ok.

Response 2: Thanks for your comment to improve the quality of the manuscript.

Comments 3: Some improvement is made.

Response 3: Thanks for your comment to improve the quality of the manuscript.

Comments 4: Ok. Although, the number of sensors described in the paper is 10, which can be done using normal FBGs. So, the advantage has not been used.

Response 4: Thanks for your comment. Yes, the advantage of weak FBGs has not been used in this paper because relatively small samples were used in laboratory. If a really large panel (more than 5 m in length) is manufactured in the industry, more sensing points will be needed and the weak FBGs can achieve the quasi-distributed flow front monitoring during the manufacturing process, which the normal FBGs can’t.

Please see the lines 588-592 of the revised manuscript marked in red.

Comments 5: I would prefer to remove the table, which has just two almost identical rows.

Response 5: Thanks for your comment. The Table 2 has been removed. The text was modified as follow:

Thanks to the Dense Wavelength Division Multiplexing technology, each OF con-sisted of ten FBGs with the grating length of 10 mm and grating distance of 50 mm. Each OF could be used as ten flow sensors with the central wavelength ranges of 1500-1600 nm of polyimide-coated OF and 1529-1569 nm of polyacrylate-coated OF.

Please see the lines 246-250 of the revised manuscript marked in red.

Comments 6: Not clear. How do you define “5 to 6 layer”?

Response 6: Thanks for your comment. A clear definition of the number of layers has been added. The sentences are modified as following:

From bottom (mold contacted with 0 layer) to top (vacuum bag contacted with 12 layer), three OFs were looped, then pasted on the mold (FBG 1,3,5,7,9 of CH1 and FBG 1,2,3,4,5 of CH2 and CH3) and embedded into 5 to 6 layer (FBG 10,8,6,4,2 of CH1 and FBG 6,7,8,9,10 of CH2 and CH3) of the fabric stack, see Fig. 2(a).

Please see the lines 256-260 of the revised manuscript marked in red.

Comments 7: Not clear. What does it mean “the gratings were made with the numbers”?

Response 7: Thanks for your comment. The Fig.3 (a) and (b) have been updated with the schematic drawing of grating numbers. And a clear definition of the number of gratings has been added: While the polyimide-coated weak FBGs were prepared in-line fabrication technology using a draw tower, the adjacent gratings were made in a spaced wavelength to avoid the mutual interference. For example, the first FBG of the polyimide-coated OF was actually the grating 2 with the central wavelength of 1542 nm, see Fig.3(a). The polyacrylate-coated OF was in the serial grating numbers from 1 to 10, see Fig.3(b).

Please see the lines 282-287 of the revised manuscript marked in red.

Comments 8: Ok.

Response 8: Thanks for your comment to improve the quality of the manuscript.

Comments 9: Ok. Check line 282. What are the number at the Y axis in Figs. 3 b,c?

Response 9: Thanks for your comment. The sentence of line 282 has been modified as following: The spectral signals of OFs of CH1, CH2 and CH3 before and after resin injection were showed in Fig. 3. The values at the Y axis in Fig.3 have been normalized and the axis have been changed into “Normalized Amplitude”.

Please see the line 281 and Figure.3 of the revised manuscript.

Comments 10: This is strange to omit some data without explanation. Every curve should have the point of resin arrival. In my opinion, it is not possible to find the flow front point from the strain curves measured by the authors if you do not know beforehand where the front is. Therefore, it is not clear how to monitor the flow front.

Response 10: Thanks for your insightful comment. All strain results have been added in Fig.4 except FBG10 of CH1, which was right in the edge of the sample and the signal was heavily disturbed. By comparing with the results monitored by the top and bottom cameras, the singular points of the strain curves were extracted and analyzed. When the resin flowed near the grating area, the vacuum pressure released partially to actuate the resin, so the stress on the fiber bed decreased. Therefore, two mechanisms including the harder compaction of wet fabric and stress relaxation will compete with each other, resulting in a complex strain curve. When the resin arrived, the factor of stress decrease was dominating and the strain began to increase, and that is also the reason the local preform thickness increases when the resin comes in the VARI process. For the bottom FBGs contacted with the nondeformable mold, the strain remained un-changed before resin arrived. The onset points where the slope began to increase sharply were defined as the time when the resin arrived, as shown in Fig. 4. Different resin arrival times could be obtained for each FBG and the curves clearly showed the sequences.

Please see the Figure.4 and lines 305-322 of the revised manuscript.

Comments 11: Eq. 4 does not have the linear term. Why do you fit the data in Fig.5 with a curve where the linear term prevails? -

Response 11: Thanks for your insightful comment. The Fig.5 has been modified without the quadratic fitting. And the monitoring results obtained by FBG array were compared with the camera: To validate the monitoring ability of FBGs array, the flow fronts monitored by the bottom camera were extracted and compared with the FBGs monitored results, as shown in Fig. 5. The good compatibilities can be found between the FBGs array and bottom camera monitored results, which showed that change onset points of the strain curves can be used to track the resin flow fronts.

Please see the Figure.5 and lines 348-355 of the revised manuscript.

Comments 12: Ok.

Response 12: Thanks for your comment to improve the quality of the manuscript.

Comments 13: Ok.

Response 13: Thanks for your comment to improve the quality of the manuscript.

Comments 14: Ok.

Response 14: Thanks for your comment to improve the quality of the manuscript.

Comments 15: Ok.

Response 15: Thanks for your comment to improve the quality of the manuscript.

Comments 16: No improvement.

Section “Discussion and conclusions” is strange. No discussions. Poor conclusions. It should be rewritten.

Response 16: Thanks for your comment. This part has been rewritten as following:

In our work, the FBGs array monitoring abilities of resin flow front and manufacturing defects in the VARI process were fully investigated. The FBG monitoring criterions based on the change onset were determined with the camera monitoring results: (a) FBGs placed directly on the rigid mold with single vacuum bag: the point when the strain began to increase; (b) FBGs placed between soft fabric with single vacuum bag: the bottom of the strain curve of “V” shape; (c) FBGs placed between soft fabric with double vacuum bags: the point when the strain began to decrease. In 2D and 3D flow monitoring tests, the results were well matched with the camera monitoring results. The changed strain could represent the manufacturing defects by comparing with the FBGs in the same conditions. Therefore, the weak FBGs array-based method is an effective way to achieve the quasi-distributed monitoring of resin flow and manufacturing defects with high sensing density.

For a general case, the algorithm to obtain the flow front points is summarized as following: 1. Determine the location of the sensor and the vacuum bagging method; 2. Choose the right monitoring criterions; 3. Analysis the strain curves to obtain the flow front points. Due to the variability and complexity of strain curves, the analysis is mainly on expert experience now and we hope to make the code to do this analysis in the future. The effect on resin flow of channel introduced by embedded OF needs to be evaluated and the compensation algorithm is a good choice for quantitative flow prediction in the manufacturing process. The method also showed the potential to experimentally characterize the local permeabilities and impregnated states under the fact that the non-uniformity of the fabric induced by the unavoided fibre distortions and nesting while stacking.

Please see the lines 579-615 of the revised manuscript.

4. Response to Comments on the Quality of English Language

Point: The English still should be improved.

Response: Thanks for your comment. We had gone through the whole text and modified some clerical and grammatical errors. And some sentences were simplified to reduce verbose descriptions.

Please see the lines 188, 193,218,224,249,272,319,347,350,357,385,434,445,481,507,554,568 of the revised manuscript.

5. Additional clarifications

There are no additional clarifications here.